# Effects of Slaughter Age on Muscle Characteristics and Meat Quality Traits of Da-Heng Meat Type Birds

**DOI:** 10.3390/ani10010069

**Published:** 2019-12-30

**Authors:** Jingjing Li, Chaowu Yang, Han Peng, Huadong Yin, Yan Wang, Yaodong Hu, Chunlin Yu, Xiaosong Jiang, Huarui Du, Qingyun Li, Yiping Liu

**Affiliations:** 1Farm Animal Genetic Resources Exploration and Innovation Key Laboratory of Sichuan Province, Sichuan Agricultural University, Chengdu Campus, Chengdu 611130, China; jingjingyi11@126.com (J.L.); yinhuadong@sicau.edu.cn (H.Y.); as519723614@163.com (Y.W.); 13982@sicau.edu.cn (Y.H.); 2Sichuan Animal Science Academy, Chengdu 610066, China; cwyang@foxmail.com (C.Y.); penghan0706@163.com (H.P.); yuchunlin1984@sina.com (C.Y.); xsjiang2017@163.com (X.J.); duhuarui@sina.com (H.D.); liqingyun2008@163.com (Q.L.); 3Animal Breeding and Genetics key Laboratory of Sichuan Province, Chengdu 610066, China

**Keywords:** chicken, slaughter age, myofiber, meat quality

## Abstract

**Simple Summary:**

The current work evaluated the breast muscle performance, meat quality traits, and myofiber characteristics of Da-Heng meat type birds with 60, 90, 120, 150, 180 days of age. Older chickens often presented a higher pH, lower drip loss, higher shear force, darker, and redder breast meat. The correlation coefficients showed that myofiber characteristics played an important role in breast pH values, drip loss, and meat color.

**Abstract:**

Due to the increasing demand for producing chickens with high meat quality, there is a need to determine its mode of action on chicken meat quality traits across a wider age spectrum. In this study, five groups of 200 male Da-Heng meat type birds were reared until slaughter age of 60, 90, 120, 150, 180 days old and breast muscle performance, meat quality traits, and myofiber characteristics were evaluated. The larger body weight and breast weight of chicken are based on larger myofiber diameter and area, less myofiber density for the older birds than younger birds. There was an age effect on all meat quality traits of chicken breast muscle (*p* < 0.05). Older chickens often presented a higher pH, lower drip loss, higher shear force, darker, and redder breast meat. The correlation coefficients showed that myofiber characteristics played an important role in breast pH values, drip loss, and meat color (*p* < 0.05). Besides, significant correlations were also found between meat quality traits (*p* < 0.05). Further studies are needed to explore the biochemical character and potential molecular mechanism of chicken breast muscle to determine the factors that causes these age-related differences in meat quality in the current study.

## 1. Introduction

China is the world’s largest meat market, consuming 18% of the chicken produced worldwide. Moreover, current forecasts and projection studies have predicted that the expansion of the poultry market will continue in the future [1]. Indeed, poultry meat, particularly breast meat, is popular due to its healthy nutritional profile, fitting the modern consumer demand for low fat, sodium, and cholesterol [2]. An intense genetic selection for fast growth rate and high breast yield has led to progressive improvements in meat production [3]. However, these advances have been often accompanied by deterioration of the meat type birds meat quality, including high incidence of muscle abnormalities, such as white striping and wooden breast [4,5]. Nowadays, chicken meat is widely sold as parts or further-processed products, indicating the importance of meat quality in the poultry industry [6]. As living standards have improved, the qualitative properties of the meat need to be guaranteed to meet the demand under quality and health conditions [7].

Several factors have been shown to influence poultry meat quality, such as breed, age, sex, slaughter weight, management, transport, slaughter procedures, aging time, etc. [7]. Much emphasis has been given to evaluate the meat quality between commercial chicken breeds and indigenous chicken breeds or fast-growing strains and slow-growing strains [8,9,10,11,12]. The indigenous chickens generally have better meat quality than commercial broiler chickens, as they may fully benefit from the slower growth rate and older age [13]. The effects of slaughter age on the meat quality characteristics have been studied on pigs, sheep, deer, emu, etc. [14,15,16,17]. However, the effects of age need be investigated across a wider age spectrum to determine its mode of action on chicken meat quality traits.

Myofiber number in chickens is established before hatching. Post-natal growth of muscle depends on the increase in length and diameter of individual myofibers [18]. It has been documented that muscle characteristics (myofiber diameter, cross sectional area, and density) could proudly influence meat quality. Several studies have shown that smaller fiber size (diameter and area), leads to tender meat [19,20,21,22]. However, a higher myofiber density might partially support a good sensorial perception of taste [23]. These studies suggested that myofiber information was an important factor for meat quality evaluation in animals. Therefore, understanding the development of the myofiber characteristics with age and its influence on the meat quality traits is instrumental in the breeding of chickens with high meat quality.

Da-Heng meat type bird is a specialized meat-type breed with stable production performance and characteristic meat quality. Using male Da-Heng meat type birds S011 line, the purpose of this study was to: (1) evaluate the effects of slaughter age on breast muscle characteristics (breast dimensions, myofiber diameter, cross sectional area, density) and meat quality traits (pH, color, water-holding capacity, shear force); (2) discuss the relationships between myofiber characteristics and meat quality traits.

## 2. Materials and Methods

### 2.1. Sample Preparation

A total of 200 1-day-old male Da-Heng meat type birds were reared in the Sichuan Da-Heng Poultry Breeding Company (Chengdu, China) with one cage for each bird. Room temperatures of 15 to 20 °C were maintained by controlled ventilation and heating. All birds entered the experiment at the same time and were kept under the same conditions with free access to feed and water at all time, following national institutes of health guide for the care and use of laboratory animals (Table 1). At the rearing period of 60, 90, 120, 150, and 180 days of age, 20 birds of the same age with similar body weights were chosen to examine the effects of age on the muscle characteristics and meat quality. The feed was withdrawn 10 h prior to slaughter but the water is available for birds. At the day of slaughter, the selected chickens were measured for body weight (BW) (g) and killed by cutting the carotid arteries, and subsequently bled for 5 min. Birds were handled in accordance with the principles and procedures outlined by Sichuan Agricultural University’s Animal Care and Use Committee.

### 2.2. Measurements of Breast Dimensions and Myofiber Characteristics

After slaughter, the breast muscles were carefully excised and weighed (g). The muscles were trimmed of obvious adipose tissue and connective tissue and then stored on ice at 4 °C until the measurements of meat quality parameters. The right breast muscle was used for the measurements of breast dimensions and myofiber characteristics. The length and width (cm) were measured at the longest and widest point of each fillet using vernier calipers. Three measurements of height (cm) were carried out at the caudal, midpoint, and cranial positions on the fillet [8]. A strip (2 × 1 × 1) from pectoralis major was cross-sectioned perpendicular to the direction of the myofibers and then immersed in a 4% paraformaldehyde solution for 24 h. The muscle sections were stained with hematoxylin and eosin (H&E) to observe the morphology of the muscle tissue. Photomicrographs were taken by a digital microscope (BA400Digital, Xiamen, China) equipped with an image analyzer (Image-Pro Plus 6.0). Pictures were magnified 400-times for calculating the diameter (μm), area (μm^2^), and density (number of myofibers per square micrometer, num./μm^2^) of the myofibers. Ten random fields were counted on each slide and the average generated.

### 2.3. Measurements of Meat Quality Parameters

The pH values at 15 min (pH_15_) and 24 h (pHu) postmortem were measured on the left breast (m. pectoralis major) using a portable pH meter (TEST0205, Shanghai, China) by penetration in breast muscle [7]. The measurements on three different areas were recorded and averaged according to the national measurement standard (NY-T/1333-2007). At 24 h postmortem, color values were measured at 3 different locations on the ventral side (bone side) of the left breast muscle that were placed on a white cutting board [8]. Color was evaluated using a Minolta colorimeter (CR-300, Konica Minolta, Japan) according to the Commission International de l’Eclairage (CIE) system, where L * represents lightness, a * redness, and b * yellowness, respectively [24]. The final value for color evaluation was the average of 3 readings.

The water-holding capacity of breast meat was estimated through drip loss and cooking loss. Drip loss was determined 4 h after slaughter. A parallelepiped meat cut weighing about 5 g was excised from the caudal part of the left fillet and hung in a covered plastic bag for 48 h at 4 °C. After storage, the cut was reweighed and the difference in weight was used to determine drip loss (%) [25]. At 24 h postmortem, the cranial part of each left fillet weighed about 35 g was sealed in a polyethylene bag and cooked in a water bath of 85 °C. Once the internal temperature of the sample reached 75 °C, the muscle samples were taken out and then cooled in crushed ice for 20 min to reach room temperature. The difference in weight before and after cooking was used to determine cooking loss (%) [8,13]. The cooled samples were then cut into strips (4 × 1.5 × 1.5 cm) parallel to the fiber axis for shearing perpendicularly to the longitudinal orientation of the myofibers using Digital Meat Tenderness Meter of Model C-LM3 (Northeast Agricultural University, Harbin, China). For each strip, 3 maximum force readings (Newtons) were taken and the average of the readings was used for data analysis. In order to keep sampling consistent, all the measurements were performed at the same anatomical positions for each sample and one person made the measurements throughout the study [8].

### 2.4. Statistical Analysis

Data were analyzed statistically by the SPSS 10.0 software (SPSS Inc., Chicago, IL, USA). The effects of age on the muscle characteristics and meat quality parameters were evaluated by using the one-way analysis of variance (ANOVA) test following a general linear model (GLM). The means of different age groups were analyzed using the Duncan’s multiple range test at the significance level of 0.05. Using age as control variable, the partial correlation coefficients were generated to evaluate the correlation of muscle characteristics and meat quality traits.

## 3. Results

### 3.1. Effects of Age on Breast Weight, Dimensions, and Myofiber Characteristics

As expected, there was an age effect (*p* < 0.05) for all variables showing increased mean values with age. The results for body live weight and breast weight are shown in Table 2. The body live weight of the chickens significantly increased from 60 to 150 days of age (*p* < 0.01), while the body live weight showed no significant differences between 150 and 180-day-groups (*p* > 0.05). However, the breast weight was extremely higher in birds at 180 days of age than that at 150 days of age (*p* < 0.01).

The slaughter age also had a significant effect on breast dimensions including length, width, and height (*p* < 0.05). Theoretically, the values of these parameters should be elevated as the age increases. The slightly reduced values of the width from 120 to 150-day-old chickens and the heights from 150 to 180-day-old chickens may be attributed to the unavoidable individual differences in the study. Similar with the effects of age on breast weight, age also had a more remarkable effect on breast dimensions of meat type birds between 60 and 90 days of age (*p* < 0.001) than between other groups (Table 3).

Concerning myofiber characteristics, the diameter of muscle fiber increased with age, as shown in Table 4 and Figure 1. Similarly, the cross sectional area of muscle fiber in meat type birds at 60 days of age (495.73 μm^2^) was much smaller than that in meat type birds at 90 (941.61 μm^2^), 120 (1083.48 μm^2^), 150 (1122.96 μm^2^), and 180 (1087.40) days of age (*p* < 0.01) (Table 4). The 60-day-old chickens had significantly higher density than chickens of the other age groups (*p* < 0.01) because the larger fibers take up more space in older birds. It is suggested that the myofibers of Da-Heng meat type birds have a faster increase or a higher rate of growth from 60 days of age to 90 days of age. Significant age effects illustrated heavier body weight, heavier breast weight, larger myofiber diameter and area, and lower myofiber density for the older birds than the younger birds.

### 3.2. Effects of Age on Meat Quality Parameters

Meat quality parameters of Da-Heng meat type birds at different slaughter ages are summarized in Table 5. The pH_15_ and pHu values measured in the breast muscle increased with the growing of age and the values at 180 days of age were remarkably higher than that at the other age stages (*p* < 0.05). The difference values between the pH_15_ and pHu reflected the decline rate in muscle pH, which presented a trend of decreasing with increasing age.

Water-holding capacity is an important meat quality attribute, which was evaluated by measuring drip loss and cooking loss in this study. The breast muscles of younger birds (day 60 and day 90) had significantly higher drip losses (*p* > 0.05) than the older birds (day 150 and day 180) (*p* < 0.05). The mean values of cooking loss in chickens at day 180 was the highest, which was significantly higher than that of 60, 90, and 150 days of age (*p* < 0.05) but showed no significant difference with that of 120 days of age. The shear force values of meat type birds at 60 and 90 days of age were significantly lower than that of meat type birds at 120, 150, and 180 days of age (*p* < 0.05).

Considering the color of breast muscles, the mean values of lightness (L *) first increased and then decreased along with the increasing of age and reached the maximum of 66.37 at 90 days of age. Interestingly, redness (a *) values was gradually elevated within the age of 60 to 150 days old, whereas the opposite trend was observed for yellowness (b *) values. Generally, the older chicken muscles exhibited darker, redder, and less yellow color than the chicken muscles with younger age.

### 3.3. Relationships between Muscle Characteristics and Meat Quality Traits

Partial correlations showed that the diameter of muscle fiber was significantly positively correlated with body live weight, breast weight, length, and width (Table 6). Similarly, the cross sectional area of myofibers was significantly positively correlated with body live weight, breast weight, and width. The density of muscle fibers was negatively correlated with breast weight, length, width, and midpoint height. However, there was no significant correlations between myofiber size and cranial height, between myofiber and caudal height of the breasts (*p* > 0.05).

The correlations between pHu and myofiber diameter and area were −0.219 and −0.212, but did not reach significant level (*p* > 0.05). The myofiber diameter was correlated with pH decline and breast meat redness (*p* < 0.05, Table 7). The density of myofibers was positively correlated with pHu (*p* < 0.05, Table 7) but negatively correlated with pH decline, drip loss and lightness of breast muscles (*p* < 0.05, Table 7).

As is shown in Table 8, there were also some significant correlations between meat quality traits. For example, pHu was positively correlated with pH_15_ (r = 0.573, *p* < 0.001), cooking loss (r = 0.29, *p* < 0.05) and negatively with pH decline (−0.425, *p* < 0.01), drip loss (−0.33, *p* < 0.05), breast meat yellowness (r = −0.569, *p* < 0.001), and shear force (r = −0.309, *p* < 0.05). Besides, the correlation between breast muscle lightness and drop loss was 0.296 (*p* < 0.05). Breast muscle redness was negatively correlated with lightness and cooking loss (r = −0.73, *p* < 0.001; r = −0.378, *p* < 0.01, respectively).

## 4. Discussion

By considering the growth performance and meat quality, our study was designed to obtain an overall assessment of the consequences of increasing slaughter age from 60 to 180 days old in broiler production. Growth intensity is important to determine the economically acceptable age for slaughter [26]. In our results, the growth intensity of Da-Heng meat type birds decreased with age. The live body weight increased by approximately 54% from 60 days to 90 days old, whereas it only increased by approximately 19% from 120 days to 180 days old. Thus, Da-Heng meat type birds at 90 days of age and 120 days of age are more appropriate to be appeared on the market considering the culture cost and growth performance.

As expected, the myofiber size, body live weight, breast weight, and dimensions increased regularly with age at slaughter. The significant positive correlations between myofiber properties and growth performance suggested that the larger body weight and breast weight of chicken were based on larger myofiber diameter and area, less myofiber density for the older birds than the younger birds, which is consistent with the earlier findings of Chen and Iwamoto et al. [18,27]. The number of myofibers does not increase after hatching, so the post-natal growth of broiler skeletal muscle was accompanied by the growth of individual myofibers [18]. Here, the age-related difference in breast muscle weight mainly resulted from the difference in fiber diameter and cross sectional area. According to Chen et al., the fiber size was the most important factor determining muscle volume [18]. Although the relationship between myofiber size and breast weight was well indicated, a small number of studies has been conducted on the role of histological properties of myofibers in the meat quality of chicken.

There was an age effect (*p* < 0.05) on all meat quality traits. Variations in the values of muscle pH and the extent of decrease in pH are responsible for variations in meat quality [28]. Low pHu (<5.8) would lead to a higher incidence of “acid meat”, with similar defects to those of PSE meat [28]. On the other hand, breast meat with high pHu (>6.0) is more likely to become contaminated with microbial growth and trigger different types of spoilage microorganisms, thus affecting storage and sensorial quality [2]. In our results, the pH levels of the breast muscles between 90 days of age and 150 days of age were appropriate to prevent PSE-like meat and improve storage quality. The older birds had higher pH15 and pHu values than younger birds in our studies, which is consistent with the previous studies. Mehaffey et al. also observed that lower muscle pH at 4 h post-mortem in birds slaughtered at 42 days compared with those birds processed at 53 days [8]. Besides, it is reported that the pH_15_ of breast muscles increased (*p* < 0.01) between 35 and 42 days of age and the pHu of breast meat increased (*p* < 0.001) between 35 and 49 days of age and then remained unchanged in a modern heavy broiler line [29]. The decline in muscle pH post-mortem is one of the most significant change that occurs during the progression of rigor mortis, which is due to the fact that the slaughtered animal glycogen is broken down into glucose [8]. Lactic acid is formed when glucose undergoes glycolysis, causing pH in muscles to drop [30]. In our results, the decline in pH of the breast muscle tended to decrease as the birds increased in age, which may lead to higher ultimate pH in older broilers. As age increased, the activity of the glycolytic enzyme phosphofructokinase (PFK) and lactate dehydrogenase (LDK) decreased in the muscles of Angus steers [31]. Baeza et al. reported that there was a concomitant decrease (*p* < 0.001) in the breast muscle lactate content with age [29]. These studies supported that the reduced glycolytic potential might be the reason for the higher pHu in older chickens. Furthermore, it is reported that the chickens with heavier slaughter weights contained higher plasma glucose than lighter chickens, suggesting the heavier chicken is more prone to pre-slaughter stress [30]. The higher sensibility before slaughter in broilers at the latter age stages could lead to poor glycogen content in muscles at the time of death [13]. Thus, it is likely that the higher pH values in older chickens also may be ascribed to their heavier carcass weights [26].

The color of meat is a further important determinant of visual appearance of consumers [12]. Since meat color can influence the purchasing decisions, it is important to characterize the change of meat color as chicken’s age increased. Brewer et al. (2001) reported that L * value was most correlated to visual color, and using L * in conjunction with a * could explain 69% of the variability in the visual pink color [32]. Our results indicated that the older chicken muscles generally exhibited darker, redder, and less yellow than the chicken muscles with younger age. Baeza et al. also reported that breast meat of broilers slaughtered at 35 days of age was lighter than that of older broilers, although they only explored the age effect from 35 days of age to 63 days of age [29]. Besides, Fletcher et al. reported that poultry breast meat tended to become darker and redder as bird’s age increased [33]. In the present study, the higher myoglobin content may contribute to the higher a * value and lower L * value in older chicken [34].

Water-holding capacity is an important meat quality trait, which is attributed to partial protein denaturation under the acidic conditions in the post-mortem muscle [2]. The values for drip loss were significantly higher in the 60-day-old and 90-day-old birds than in the 150-day-old and 180-day-old birds, while the values for cooking loss of younger birds were significantly lower (*p* < 0.05). The different changes for drip loss and cooking loss suggested the different water-holding capacity in the raw meat and processed meat products. Tenderness is regarded as the most important factor in the consumer’s perception of meat and influences consumers’ ultimate satisfaction of the poultry meat products [33]. The shear force values were used to assess the tenderness of cooked meat products. In the present investigation, the mean shear force values for breasts were age-dependent, with higher shear force values in older birds. The age-related findings in this study were consistent with those of Chen and Fletcher et al. [18,33]. Collagen is an abundant connective tissue protein. The content and the solubility of it were closely related to meat tenderness [21]. According to the earlier researches, the differences in the physiological maturity of the chickens at the time of slaughter resulted in the difference in collagen cross-linking [35]. It was noticed that the quantity of collagen increased and the solubility decreased with the age of the animals, which could result in the decrease of tenderness in older chickens [34].

As already mentioned above, increased slaughter age significantly increased muscle fiber size. Some researchers suggested that larger fiber size was often associated with higher shear force and meat toughening [12,18]. An increase in fiber size may represent a change in the myofibrillar portion and thus impact the textural properties of meat [36]. This change was due to the muscle metabolism at the time of slaughter [23]. In this study, the larger myofiber size was indeed positively associated with shear force while no significance was observed (*p* > 0.05). According to Chen et al., it is likely that larger diameter of myofibers resulted in higher shear force, partly due to greater thickness of the perimysium of the muscles [18]. Moreover, some researches indicated that chicken breast muscles exhibiting larger fiber size exhibited higher pH15 and pHu values, produced breast meat with reduced L * and lower drip loss [37], which is inconsistent with our study. Mehaffey indicated that the meat quality traits showed significant differences among different breeds in most cases and the relations within these traits were not consistent [8]. Thus, the contrasting data reported in previous studies and this study might be partially due to the different breed analyzed, which greatly influenced the characteristic of the meat.

In this study, we also evaluated the relationships between meat quality traits. The importance of pH for breast meat quality in poultry has been widely discussed. Most studies suggested that poultry meat with a low pHu value was highly associated with low water-holding capacity [12,38]. However, Sandercock et al. indicated that breast muscles with apparently lower pHu values exhibited a lower degree of drip loss, which could result in higher water-holding capacity [39]. In the present study, chicken muscle with a higher pHu value was closely related to a lower drip loss but a higher cooking loss, which resulted in the higher water-holding capacity in raw meat and lower water-holding capacity in cooked meat, respectively. The different results in drip and cooking loss suggested the changes of water-holding capacity during the processing. Thus, a further mechanism study is needed to clarify the relationship between muscle pH and water-holding capacity including drip loss and cooking loss. Besides, studies had previously shown that muscle pH and meat color were highly correlated [6,40,41,42]. Swatland et al. and Debut et al. demonstrated that a high pHu value was associated with a low L * value due to the weak light scattering in high-pH chicken meat [6,40]. Qiao et al. observed a positive correlation between pH and redness (a *) values [42]. However, our study has not led to similar conclusions. No significant correlations were found between muscle pH values and L * or a * but both pH15 and pHu showed strongly negative correlations with b *. The drip loss and cooking loss were positively associated with shear force, although no significance was observed. High amounts of liquid losses are usually associated with increased toughness of meat [43].

## 5. Conclusions

In conclusion, we have demonstrated that Da-Heng meat type birds at 90 days of age are appropriate to be on the market comprehensively considering the culture cost, growth performance, and meat quality. Older chickens often present a higher pH, lower drip loss, higher shear force, darker, and redder breast meat. The correlation coefficients obtained in this experiment showed that histological properties of myofibers plays an important role in breast pH values, drip loss, and meat color.

## Figures and Tables

**Figure 1 animals-10-00069-f001:**
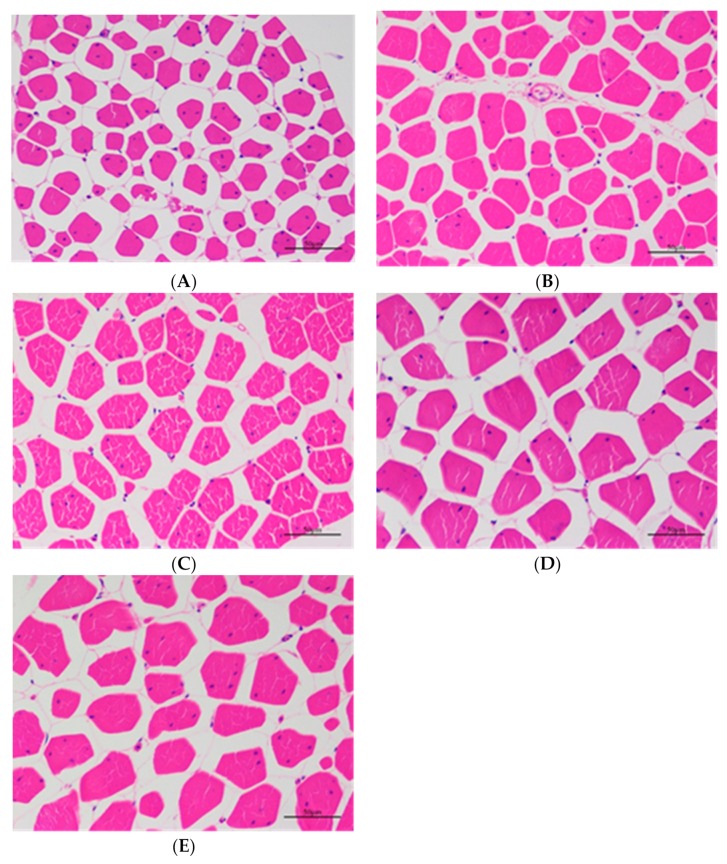
Histological evaluation of muscle fibers in samples with different slaughter age (400×). (**A**) 60 days. (**B**) 90 days. (**C**) 120 days. (**D**) 150 days. (**E**) 180 days.

**Table 1 animals-10-00069-t001:** Composition of diets distributed during the rearing period for Da-Heng meat type birds.

Ingredient (%)	Starter (1–42 d)	Grower (42–133 d)	Finisher (133–147 d)	Withdrawal (147–180 d)
Corn	59.98	58.55	60.04	61.45
Soybean meal	22.49	12.49	18.99	20.98
Rape seed cake	2.50	-	-	-
Bran	-	3.00	3.00	-
Distillers Dried Grains with Soluble	2.50	4.00	-	-
Corn germ meal	8.00	17.79	11.29	7.49
Soybean oil	0.80	0.50	0.50	0.80
Limestone (Roughness)	-	-	2.50	4.00
Limestone	1.80	1.80	2.00	3.50
CaHPO4	1.00	1.00	0.80	0.90
NaCl	0.30	0.30	0.30	0.30
L-Lys	0.16	0.18	0.10	0.10
DL-Met	0.12	0.07	0.15	0.15
Choline chloride (60%)	0.10	0.10	0.10	0.10
Multidimensional	0.04	0.03	0.04	0.04
Mineral addition	0.15	0.15	0.15	0.15
phytase	0.02	0.02	0.02	0.02
Compound enzymes	0.02	0.02	0.02	0.02
Zinc bacitracin	0.02	-	-	-
Nutrient composition				
ME(MJ/kg)	11.75	11.32	11.32	11.32
CP (%)	18.30	15.30	16.00	16.00
Calcium (%)	1.00	0.97	1.96	3.12
Total phosphorus (%)	0.60	0.65	0.56	0.53
Non-phytic acid phosphor	0.30	0.29	0.27	0.27
Crude fiber (%)	2.93	3.04	2.70	2.40
Lysine (%)	0.96	0.75	0.83	0.84
Methionine (%)	0.41	0.31	0.40	0.40

**Table 2 animals-10-00069-t002:** Effect of age on growth performance of Da-Heng meat type birds.

Traits	Age				
	60	90	120	150	180
Body live weight (g)	1178.00 ^a^ ± 24.72	1816.00 ^b^ ± 47.18	2472.50 ^c^ ± 50.87	2844.75 ^d^ ± 63.87	2943.75 ^d^ ± 57.72
Breast weight (g)	90.61 ^a^ ± 2.55	188.54 ^b^ ± 6.42	251.22 ^c^ ± 5.86	298.90 ^d^ ± 10.72	327.32 ^e^ ± 8.39

Values are expressed as mean ± standard error (SE). ^a–e^ Means within rows with different superscript letters differ significantly (*p* ≤ 0.05).

**Table 3 animals-10-00069-t003:** Effect of age on breast dimensions of Da-Heng meat type birds.

Traits	Age				
	60	90	120	150	180
Length, cm	135.65 ^a^ ± 2.71	163.29 ^b^ ± 2.48	172.94 ^c^ ± 2.49	179.83 ^cd^ ± 2.92	184.10 ^d^ ± 2.67
Width, cm	45.48 ^a^ ± 1.61	63.76 ^b^ ± 1.19	73.55 ^cd^ ± 1.27	71.70 ^c^ ± 1.41	76.75 ^d^ ± 1.21
Cranial height, cm	6.09 ^a^ ± 0.46	10.91 ^b^ ± 0.54	12.54 ^b^ ± 0.46	16.51 ^c^ ± 0.76	16.54 ^c^ ± 0.74
Midpoint height, cm	9.44 ^a^ ± 0.43	14.87 ^b^ ± 0.68	14.88 ^b^ ± 0.66	18.38 ^c^ ± 0.65	17.66 ^c^ ± 0.64
Caudal height, cm	3.62 ^a^ ± 0.24	7.75 ^b^ ± 0.35	8.69 ^b^ ± 0.29	11.88 ^c^ ± 0.56	11.74 ^c^ ± 0.50

Values are expressed as mean ± SE. ^a–d^ Means within rows with different superscript letters differ significantly (*p* ≤ 0.05).

**Table 4 animals-10-00069-t004:** Effect of age on myofiber characteristics of Da-Heng meat type birds.

Trait	Age				
	60	90	120	150	180
Diameter (μm)	24.50 ^a^ ± 0.67	33.44 ^b^ ± 1.02	35.67 ^b^ ± 2.00	35.86 ^b^ ± 2.55	35.74 ^b^ ± 1.02
Cross sectional area (μm^2^)	495.73 ^a^ ± 27.21	941.61 ^b^ ± 57.51	1083.48 ^b^ ± 110.67	1122.96 ^b^ ± 163.06	1087.40 ^b^ ± 56.32
Density (num/μm^2^)	14.46 ^a^ ± 1.20	6.78 ^b^ ± 0.72	6.28 ^b^ ± 0.66	6.18 ^b^ ± 0.50	6.11 ^b^ ± 0.24

Values are expressed as mean ± SE. ^a,b^ Means within rows with different superscript letters differ significantly (*p* ≤ 0.05).

**Table 5 animals-10-00069-t005:** Effect of age on meat quality traits of Da-Heng meat type birds.

Trait		Age				
		60	90	120	150	180
pH_15_		5.94 ^a^ ± 0.035	6.12 ^b^ ± 0.036	6.17 ^bc^ ± 0.031	6.21 ^c^ ± 0.015	6.69 ^d^ ± 0.014
pHu		5.66 ^a^ ± 0.025	5.81 ^b^ ± 0.034	5.89 ^c^ ± 0.025	6.02 ^d^ ± 0.017	6.58 ^e^ ± 0.0087
pH decline		0.28 ^a^ ± 0.031	0.31 ^b^ ± 0.034	0.28 ^b^ ± 0.032	0.19 ^b^ ± 0.021	0.10 ^c^ ± 0.019
Drip loss (%)		3.93 ^a^ ± 0.35	4.09 ^a^ ± 0.39	3.57 ^ab^ ± 0.29	2.96 ^b^ ± 0.22	2.73 ^b^ ± 0.28
Cooking loss (%)		18.78 ^a^ ± 0.73	18.82 ^a^ ± 0.74	21.71 ^b^ ± 0.70	19.48 ^a^ ± 0.48	23.28 ^b^ ± 0.70
Shear force (N)		19.25 ^a^ ± 0.74	24.03 ^a^ ± 1.17	37.21 ^b^ ± 3.63	37.43 ^b^ ± 3.89	38.93 ^b^ ± 4.94
Color	Lightness (L ^*^)	62.27 ^a^ ± 0.60	66.37 ^b^ ± 0.83	64.96 ^b^ ± 0.71	60.27 ^a^ ± 0.92	60.61 ^a^ ± 0.95
	Redness (a ^*^)	8.32 ^a^ ± 0.28	8.90 ^a^ ± 0.29	9.00 ^a^ ± 0.45	12.28 ^b^ ± 0.35	11.37 ^b^ ± 0.69
	Yellowness (b ^*^)	16.82 ^a^ ± 0.33	11.98 ^b^ ± 0.49	11.59 ^bc^ ± 0.38	10.51 ^c^ ± 0.51	8.01 ^d^ ± 0.67

Values are expressed as mean ± SE. ^a–d^ Means within rows with different superscript letters differ significantly (*p* ≤ 0.05).

**Table 6 animals-10-00069-t006:** Partial correlation coefficients between breast performance and myofiber characteristics for Da-Heng meat type birds.

Variable	Age	Body Live Weight	Breast Weight	Length	Width	Cranial Height	Midpoint Height	Caudal Height	Diameter	Cross Sectional Area
Diameter	0.454 **	0.454 **	0.584 ***	0.386 **	0.503 ***	0.162	0.345 *	0.154		
Cross sectional area	0.424 **	0.446 **	0.569 ***	0.371 *	0.476 **	0.186	0.332 *	0.173	0.991 ***	
Density	−0.532 ***	−0.326 *	−0.494 ***	−0.375 **	−0.51 ***	−0.168	−0.42 **	−0.204	−0.671 ***	−0.621 ***

The superscript asterisk (*), double asterisk (**), treble asterisk (***) denotes statistical difference of *p* < 0.05, *p* < 0.01, *p* < 0.001, respectively.

**Table 7 animals-10-00069-t007:** Partial correlation coefficients between myofiber characteristics and meat quality traits.

Variances	pH_15_	pHu	pH Decline	Drip Loss	Cooking Loss	Shear Force	L *	a *	b *
Diameter	0.055	−0.219	0.29 *	0.154	−0.081	0.166	0.217	−0.289 *	−0.162
Cross sectional area	0.039	−0.212	0.263	0.117	−0.07	0.151	0.18	−0.27	−0.131
Density	0.035	0.348 *	−0.334 *	−0.447 **	0.177	−0.154	−0.339 *	0.203	0.219

The superscript asterisk (*), double asterisk (**) denotes statistical difference of *p* < 0.05, *p* < 0.01, respectively.

**Table 8 animals-10-00069-t008:** Partial correlation coefficients between meat quality traits for Da-Heng meat type birdss.

Variances	Age	pH_15_	pHu	pH Decline	Drip Loss	Cooking Loss	L *	a *	b *
pH_15_	0.761 ***								
pHu	0.876 ***	0.573 ***							
pH decline	−0.53 ***	0.498 ***	−0.425 **						
Drip loss	−0.43 **	-0.095	−0.33 *	0.248					
Cooking loss	0.436 **	0.082	0.29 *	−0.216	−0.019				
L *	−0.295 *	0.207	−0.019	0.255	0.296 *	0.22			
a *	0.579 ***	−0.203	−0.089	−0.135	−0.135	−0.378 **	−0.73 ***		
b *	−0.754 ***	−0.484 **	−0.569 ***	0.066	0.003	−0.196	−0.305 *	0.262	
Shear force	0.522 ***	−0.309 *	−0.309 *	0.104	0.247	0.166	0.006	−0.196	0.136

The superscript asterisk (*), double asterisk (**), treble asterisk (***) denotes statistical difference of *p* < 0.05, *p* < 0.01, *p* < 0.001, respectively.

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
