# Peer review of "Effects of Slaughter Age on Muscle Characteristics and Meat Quality Traits of Da-Heng Meat Type Birds"

_animals, 2019, doi:10.3390/ani10010069_

Round 1

Reviewer 1 Report

Please find attached Review.

Author Response

Reviewer 1

Poultry meat quality has been widely studied because of a growing demand of the market on high broiler chicken meat quality mainly due to increasing consumer awareness of importance of animal welfare and meat quality. Presented results showed the relationships between broiler slaughter age, meat quality traits and fiber characteristics. The research results obtained by the authors fit in the current research on poultry meat quality.

Manuscript presents an interesting research results, experiment was properly designed and methods were appropriately match. However, I have some suggestions and minor corrections.

Line number:

17-Abstract: please put here more details, statistical importance of results and add conclusions.

Response>: Thanks for your suggestions. We have added more information in the abstract.

86-Material and methods: please give units of a strip you cut to analysis.

Response>: Thanks for your suggestions. We have added the units of the traits in the part of material and method.

119-Stastistical analysis: write that Pearson correlation was used-it appears in the section “Results” but it will be better to write it earlier.

Response>: Sorry for our mistake. In this study, partial correlation coefficients were generated with age as control variable. We have changed “Pearson correlation” into “Partial correlation” in the part of results.

138-Please add a p-value to all tables. The values in table should be more readable-maybe small space between results and mean SE or different font size, bigger for results values and smaller one for mean SE?

Response>: Thanks for your suggestions. As the p-value was generated between any two groups, it is difficult to present every p-value between any two groups in the table. Thus, the superscripts letter was used to show the significant level of 0.05. Besides, we have revised the table with bigger for results values and smaller one for mean SE.

339-Conclisuions: it is summery of research rather than conclusion. It should contain the most important conclusions. Quality traits of meat which are important to producers of broiler meat may be different to those of the processor and consumer…

For example: conclude what slaughter age is the best taking into consideration producers or processors and consumer preference of meat quality? The beat meat quality traits was achieve at slaughter age…

Response>: Thanks for your suggestions. We have deleted some unnecessary conclusions and demonstrated that Da-Heng broilers at 90 days of age is appropriate to be appeared on the market comprehensively considering the culture cost, growth performance and meat quality in the conclusion.

Reviewer 2 Report

Review attached.

Author Response

The manuscript addresses a topical issue of the important factors influencing of meat quality in poultry. The study expands the existing knowledge base and provides new insights into relationships myofiber characteristics and meat quality traits of chickens, in particular DaHeng broiler.
The rationale and objective of the study are clearly stated, and the methods are appropriate. It is suggested to introduce certain corrections:
1) The information on bird nutrition should be expanded. Nutrition is a very important factor shaping the quality of meat, so such data should be placed, e.g. in the form of a table: Composition of the diet.

Response>: Thanks for your suggestion. We have added a table presenting composition of diets distributed during the rearing period for Da-Heng broilers.

2) The description of the results provided information on significance at the level of p<0.01, in relation to the effect of the experimental factor (age), however on the tables did not provide p-value. Can Authors explain why ?
Example below:
“…..The results for body live weight, breast weight and breast yield are shown in Table 1. The body live weight of the chickens significantly increased from 60 to 150 days of age (p<0.01), while the body live weight showed no significant differences between 150 and 180-day-groups (p>0.05). However, the breast weight was extremely higher in birds at 180 days of age than that at 150 days of age (p<0.01), which explains why the breast yield is significantly higher in 180-day-group than 150- day-group (p<0.01).
There were extremely lower breast yield values in the chickens at 60 days of age compared with the chickens at the other age stages (p<0.01). However, no significant differences were observed in 90, 120 and 150-day-groups (p>0.05), indicating that the breast yield verged to stabilization after chickens reached 90 days of age….”

Response>: Using the one-way analysis of variance (ANOVA) test in the SPSS 10.0, we could obtain the specific p-value between any two groups. Thus, the description of the results provided information on significance at the level of p<0.01. However, it is clearer to present the results with the letter superscripts at the level of p<0.05 than provide the p-value in the table. 

3) The conclusion "In conclusion, we have demonstrated that increasing slaughter age is an efficient way to produce heavy broilers and improve meat production" is not appropriate. It should be clarified. This effect applies to Da-Heng broilers, but it is difficult to generalize.

Response>: Thanks for your correction and we have corrected it in the part of conclusion.

4) Simple Summary- It should be added that the study was conducted on Da-Heng broilers, as this may mislead an inexperienced reader, because the commercial broiler chickens are slaughtered at the age of 35-42 days old.

Response>: Thanks for your suggestions and we have corrected it in the part of simple summery.

Yours faith fully
Reviewer

Reviewer 3 Report

all comments are in the attached file

Author Response

Line 60- Da-Heng is a commercial hybrid or a indigenous breed? Typical broilers are fast-growing birds(daily≈60g) reared for relatively short period (35-42 days). In this study, the birds were kept for a long time but reached a low final weight. It is suggested to write “meat type birds” instead of “broiler”.

Response>: Thanks for your suggestions. Da-Heng is a cultivated breed by Sichuan Da-Heng Poultry Breeding Company. We have changed “broiler” into “meat type birds” throughout the manuscript.

Line 69- Why were birds kept in metabolic cages? It is a strict laboratory environment and it is impossible to compare the results to standard litter farm. What was the size of the cage? Did the birds have ability to move?

Response>: Following national institutes of health guide for the care and use of laboratory animals, cage provides exceed 700cm2 space for each bird, with enough space to move. Each cage group provides enough space for natural mating of 3 male chickens and 24 female chickens.

Line 74- It is necessary to provide the basic composition of the feed, without it it is impossible to conclude if low final body weight results from low nutrients concentration in the feed mixture or rather from small growth rate of this particular breed.

Response>: Thanks for your suggestions. We have provided the Composition of diets distributed during the rearing period for Da-Heng meat type birds in table 1.

Line 81-Ice cooling did not change the color of the pectoral muscle?

Response>: The fillet was packaged and stored on ice until 24 h postmortem at 4°C. After 24 h, the muscle converted to mature meat and the color become stable. The method of measurements refers to Mehaffey et al. (Mehaffey, J. M.; Pradhan, S. P.; Meullenet, J. F.; Emmert, J. L.; Mckee, S. R.; Owens, C. M. Meat quality evaluation of minimally aged broiler breast fillets from five commercial genetic strains. Poultry science 2006, 85, 902-908.)

Line 101-“Higher L*, a*…[24]”-if the scale is given this sentence is unnecessary.

Response>: Thanks for your suggestions and we have deleted it.

Line 105-why samples for the drip loss were so small?

Response>: The small drip loss may be due to the type of broiler used.

Table 1- Why the proportion of breast muscle was calculated in relation to the live body weight of birds? This is indicated by the data in the table. Generally, the carcass yield is calculated like this, but the proportions of particular elements are estimated as parts of the carcass. So the carcass weight should be added or breast proportion should be rather “≤0.05” than “<0.05”, according to the statistical principles

Response>: Sorry for our mistake. Unfortunately, the chickens were killed but we did not collect the data of the carcass weight. We could not provide the carcass weight and we have deleted the information of breast yield in the table 1. We will pay more attention to the detail of methods in future experiments. Besides, we have changed “<0.05” into  “≤0.05”. Thanks for your suggestions.

Figure 1-There is no need to add photos, data are listed in Table 2. They do not bring any relevant information to the article.

Response>: Thanks for your suggestions. We have deleted the figure 1.

Line 160- The table data should not be literally repeated in the text. Generally, authors should avoid such repetitions.

Response>: Thanks for your suggestions and we have deleted some unnecessary repetitions.

Figure 2- The description(legend) should be simplified e.g. Histological evaluation of muscle fibers stained H&E in samples with different slaughter age (400×) (A)60days etc. All samples were stained by the same method so it is not necessary to repeat it so many times.

Response>: Thanks for your suggestions and we have simplified the description of figure 2.

Line 177-All used abbreviations should be explained when they first time appear in the text (these in the M&M section)

Response>: Thanks for your suggestions. We have explained the abbreviations in the part of materials and methods

Line 179- “Breast muscle pHu…”it is not description of results but part of M&M

Response>: Thanks for your suggestions. We have deleted this sentence.

Line 180- All abbreviations should be unified (pH)

Response>: Thanks for your suggestions. We have checked it again and unified all abbreviations.

Line 190- the possible reason of the difference should be explained

Response>: It is explained in the part of discussion.

Line 200- There is no need to repeat the data from the table

Response>: Thanks for your suggestions. We have deleted the repeated data in the revised manuscript.

Table 5- in 1st column should be rather “variable” instead of “variance”. The table should be corrected so that the text fits in the columns, i.e. the last column may be deleted, there is no data. Also there is no need to mark the lack of statistical significant of coefficients. The table will be more clear.

Response>: Thanks for your suggestions. We have corrected it.

Table 6- the same comments as to table 5

Response>: Thanks for your suggestions. We have corrected it.

Table 7- the same comments as to table 5 and 6

Response>: Thanks for your suggestions. We have corrected it.

Discussion- The discussion is concise, although it should explain also 1st part of the study purpose (refer to the results for other meat type chickens) in terms of body weight and meat quality. At this moment it is also focused on the relationships between meat traits

Response>: Thanks for your suggestions. We explained 1st part of the study purpose in the second paragraph in the part of discussion.

Line 288- “It was evaluated by…”-it is repetition

Response>: Thanks for your suggestion and we have deleted this sentence.

Line 291- probably not different water holding capacity depending on the bird age, but the various amount intramuscular fat which is accumulated with age. It is confirmed by bigger cooking loss and shear force.

Response>: We did not imply that different water holding capacity depended on the bird age. We suggested the different water-holding capacity in the raw meat and processed meat products.

Line 293- creatine kinase activity was not evaluated in this study, it may be omitted

Response>: Thanks for your suggestion. We have deleted it.

Conclusions-presented conclusions are rather a summery of the results, and the opinion that the assessment of factors affecting the meat quality characteristics is very lactonic and is not a conclusion in the strict sense

Response>: Thanks for your suggestions and we have deleted some unnecessary content in the part of conclusion.

There is a lot of linguistic mistake i.e.:

Line 54- the smaller fiber size

Line 112-“stripes” instead of “strip”

Line 115-word “to” should be deleted (to were taken…)

Line 183- it should be rather “Mean within rows with different superscript letters differ significantly”

Line 186- the repetition “in the study” should be deleted

(It is recommender to check the text in term of their elimination.)

Response>: Thanks for your suggestion. We have checked these mistakes again and corrected them.

General recommendation

The work contains many inaccuracies and raised doubts i.e. inability to link obtained results with the production conditions or farm practice. The article needs the major revision.

Round 2

Reviewer 3 Report

the article may be pubished in present form